environmental chemistry

chemical modification, pomelo peel's pulp, adsorbent, uranyl ions, adsorption

**Author for correspondence:**
Pengfei Yang
e-mail: ypengfei2008@126.com

This article has been edited by the Royal Society of Chemistry, including the commissioning, peer review process and editorial aspects up to the point of acceptance.

# Preparation of modified pomelo peel's pulp adsorbent and its adsorption to uranyl ions

Pengfei Yang[1,2,3], Yuanhe Xu[1], Jie Tuo[1], Ang Li[1], LixiAng Liu[1] and Huiting Shi[1]

[1]School of Chemistry and Chemical Engineering, [2]Hunan Province Engineering Research Center of Radioactive Control Technology in Uranium Mining and Metallurgy, and [3]Hunan Province Engineering Technology Research Center of Uranium Tailings Treatment Technology, University of South China, Hengyang, Hunan 421001, People's Republic of China

PY, 0000-0002-8979-0437

Using pomelo peel's pulp (PPP) as raw material, a new chemically modified PPP was prepared by the process of fermentation, cooking, freeze-drying, and so on. The adsorbent has been characterized by EDS, IR, BET and SEM. The factors of different adsorption conditions such as pH value, adsorption temperature, mass of adsorbent, adsorption time and initial concentration of $UO_2^{2+}$ were investigated. The adsorption mechanism was explored by adsorption thermodynamics and kinetics experiments. The results indicate that the pH value is 6.0, the dosage of adsorbent is $500\ mg\ l^{-1}$, the temperature is $50°C$ and the adsorption time is $90\ min$, which is the best adsorption condition. When the initial concentration of $UO_2^{2+}$ is $35\ mg\ l^{-1}$, the adsorbed amount of uranyl ions by the modified PPP adsorbent can reach $42.733\ mg\ g^{-1}$, 26.8% higher than the adsorption amount of unmodified adsorbent ($31.276\ mg\ g^{-1}$), which is obviously enhanced. The kinetic and thermodynamic experiments show that the adsorption process is in good agreement with the pseudo second-order kinetics model, that it is an endothermic reaction, and the reaction is spontaneous. The adsorption process is entropy-dominated. The Freundlich adsorption isotherm can describe the adsorption process more accurately.

## 1. Introduction

With the continuous development of the nuclear industry, a large amount of uranium-containing wastewater has been generated, causing great harm to the human body and the environment [1,2]. How to effectively treat uranium-containing wastewater

has become the current research hotspot. At present, there are mainly methods for treating wastewater containing uranium: membrane separation method [3], adsorption method [4–12], chemical precipitation method [13] and electrolysis method [14], of which the adsorption method is the most efficient and simplest method [15–19]. This caused widespread concern among scholars at home and abroad. Tao et al. [20] used barium metaphosphate as an adsorbent to study the adsorption properties of uranyl ions under various conditions, and the maximum adsorption capacity could reach 261.78 mg g$^{-1}$ under the optimal conditions; Zong-bo et al. [21] synthesized polyamino-containing ligands modified ordered mesoporous materials with surface area 1052 cm$^2$ g$^{-1}$, and the adsorption of uranium could reach 454 mg g$^{-1}$. Although the above studies obtain excellent results on adsorption of uranyl ions, they used chemical synthesis to prepare materials, and raw materials are not easily available.

Pomelo production is extremely rich in China. PPP is a by-product of pomelo. A large number of pomelo peels are discarded each year, polluting the environment and wasting resources. PPP has a rough surface, a large pore size, and good adsorption [22]. PPP is expected to become a potential adsorbent. Pei et al. [23] used the ZnCl$_2$ activation method to modify citron skin to prepare absorbent, and the adsorption rate of Pb$^{2+}$ could reach >90% under optimal conditions; Yanhua et al. [24] studied different kinds of modified grapefruit skin: the results showed that the alkalized modified grapefruit skin adsorbent had the best effect, and that the adsorption rate could reach >92% under suitable conditions for Pb$^{2+}$. The above research indicates that the modified biomass material is a better potential adsorbent. Therefore, the modified PPP adsorbent was prepared, and this work not only develops a new route for the recycling of PPP, but also an efficient means for removal of UO$_2^{2+}$ in water, which would represent a high application value.

In this study, a new chemically modified PPP adsorbent was prepared, and the adsorbent has been characterized by EDS, IR, BET and SEM. The factors of different conditions such as pH value, temperature, mass of adsorbent, adsorption time, and initial concentration of UO$_2^{2+}$ on adsorption performance were also investigated. The adsorption mechanism was explored through adsorption thermodynamics and adsorption kinetics experiments.

# 2. Material and methods

## 2.1. Reagents

Sodium thiosulfate, starch, AR (Tianjin sailboat Chemical Reagent Technology Co., Ltd.); methanol, acetic acid, AR (Hunan Huihong Reagent Co., Ltd.); azo II isopropyl formate (Jiuding Chemicals Co., Ltd.); oxaloacetic acid (Beijing Balingway Technology Co., Ltd.), U$_3$O$_8$, AR (Beijing Nuclear Industry Institute of Chemical Metallurgy); NaOH, AR (Tianjin Aobokai Chemical Co., Ltd.); hydrochloric acid, nitric acid, AR (Hengyang Kaixin Chemical Reagent Co., Ltd.); anhydrous sodium acetate, AR (Guangdong Provincial Chemical Reagent Engineering Research and Development Center); chloroacetic Acid, AR (Tianjin Fuchen Chemical Reagent Co., Ltd.); and azoarsenic III, AR (Tianjin Kemiou Chemical Reagent Co., Ltd.).

## 2.2. Instrumentation

X-Max EDS meter (Oxford); Micro for TriStar II Plus 2.02 BET tester (Micro); JSM-7500F SEM analyzer (JEOL); Nicolet-460 Fourier transform infrared spectrometer (Thermo Fisher Scientific, USA); FA2004 electronic balance, accuracy of 0.0001 g (Shanghai Haoyu Hengping Scientific Instrument Co., Ltd.); T6Xinyue visible spectrophotometer (Beijing General Analysis Instrument Co., Ltd.); 6219 pH meter (Shanghai Renshi Electronics Co., Ltd.); 101-1BS-Electric blast drying box (Bangxi Instrument Technology (Shanghai) Co., Ltd.); TD5A-WS desktop low-speed centrifuge (Changsha Well Kangxiang Eagle Centrifuge Co., Ltd.); LGJ-18 freeze drying machine (Ningbo Xinzhi Biotechnology Co., Ltd.); KQ2200DE Chinese liquid crystal ultrasonic cleaner (Kunshan Meimei Ultrasonic Instruments); and DF-101S collector-type thermostatically heated magnetic stirrer (Changsha Taikang Instrument Equipment Co., Ltd.).

## 2.3. Preparation of adsorbent

Pomelos purchased on the market were stripped to obtain PPP materials. After the PPP was shredded, soaked in deionized water and a small amount of Na$_2$S$_2$O$_3$ was added to keep the dissolved oxygen in the water at about 0.4 mg l$^{-1}$. Then a particular amount of methanol and acetic acid solution was added.

After maintaining the above state for 30 days, the PPP was taken out and mixed with starch at a mass ratio of 10 : 1. Next, a particular amount of 0.3% isopropyl azodicarboxylic acid was added and stirred for 24 h. PPP was placed on the upper layer, and 20% oxaloacetate solution was placed on the lower layer, which was boiled in the autoclave for 60 min. The sample was then cooled and dried in a vacuum freeze dryer to obtain a fluffy white adsorbent.

## 2.4. Determination of U(VI)

In total, 0.5 ml of the solution under testing and 0.5 ml arsenazo III solution were added to a 5 ml volumetric flask. The volume was adjusted with sodium acetate-chloroacetate buffer solution, the mixture was shaken well and stirred for 25 min. The absorbance of the solution was measured using a spectrophotometer at 652 nm wavelength with a blank reagent as a reference.

## 2.5. Adsorption experiment

Twenty ml of uranyl ion solution was added to a 50 ml beaker to change the adsorption experiment conditions: pH value, temperature, mass of sorbent, adsorption time, the initial concentration of uranyl ion, etc. All experiments were conducted in a thermostatically heated magnetic stirrer. The adsorbed solution was centrifuged. After separation, the uranyl ion concentration in the supernatant was measured by spectrophotometry.

The adsorption efficiency ($\theta$) and the adsorption capacity ($q$) are calculated as follows:

$$\theta = \frac{C_0 - C_e}{C_0} \times 100\% \tag{2.1}$$

and

$$q_t = \frac{C_0 - C_e}{m} \times V, \tag{2.2}$$

where $\theta$ is the adsorption efficiency (%), $q_t$ is the adsorption amount at time $t$ (mg g$^{-1}$), $C_0$ is the uranyl ion initial concentration (mg l$^{-1}$), $C_e$ is the uranyl ion equilibrium concentration (mg l$^{-1}$), $m$ is the mass of sorbent (g), and $V$ is the volume of the solution (l).

# 3. Results and discussion

## 3.1. Standard curve

Different concentrations of uranyl ion solution were prepared to the mark line in a series of 5 ml volumetric flasks, and the absorbance of uranyl ion solutions was measured. The standard curve of the uranium ion concentration and absorbance was plotted in figure 1

$$A = 0.02217C - 0.00426, \tag{3.1}$$

where $A$ is absorbance and $C$ is uranyl ion concentration (mg l$^{-1}$).

Fitting the absorbance to the uranyl ion concentration as a function of (equation (3.1)) $R^2 = 0.9998$, which proves a good fit.

## 3.2. BET

The specific surface area, average pore diameters and pore volume of the modified PPP measured by the BET method are listed in table 1. According to the data in the table, the BET surface area of the unmodified PPP and the modified PPP is 1.061 and 36.733 m$^2$ g$^{-1}$, respectively, indicating that the modified PPP adsorbent with a large BET surface area has potential adsorption properties for UO$_2^{2+}$.

## 3.3. EDS

EDS analysis of modified PPP and adsorbent after adsorption of uranyl ion (AAUI) was performed, and the results are shown in table 2 and figure 2. According to the data in the table, the modified PPP contains mainly C and O, and it also contains a small amount of K, Ca and other elements. The mass

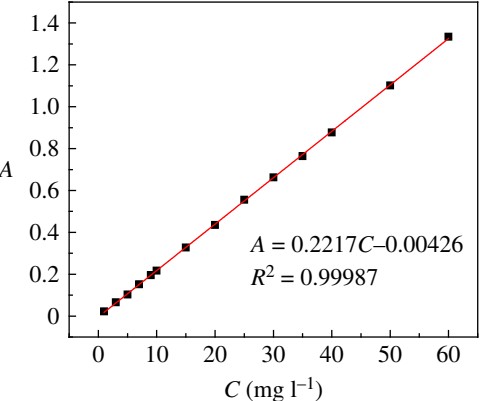

**Figure 1.** The standard curve of uranyl ion concentration versus absorbance.

**Table 1.** BET surface area, pore volume and average pore diameters of samples.

| samples | BET surface area ($m^2 g^{-1}$) | pore volume ($cm^3 g^{-1}$) | pore diameters (nm) |
|---|---|---|---|
| unmodified PPP | 1.061 | 0.0021 | 1.450 |
| modified PPP | 36.733 | 0.0336 | 3.656 |

percentage of U in the adsorbent before and after adsorbing uranium increased from 0 to 8.96%, and the atomic percentage increased from 0 to 0.57%, which shows that a large amount of uranium ions are adsorbed on the modified PPP.

### 3.4. SEM

The modified PPP and AAUI were analysed by SEM, and the results are shown in figure 3. It can be seen from figure 3 that the surface of the modified PPP has good pore structure, rough surface and a continuous and uniform porous structure, which is favourable for adsorption. The surface porosity of PPP is reduced after adsorption of uranyl ion, and the overall look of its surface is changed to be relatively flat, which is caused by the adsorption of uranyl ions.

### 3.5. IR

The infrared spectra of the modified PPP and AAUI are shown in figure 4. It can be seen from figure 4a that the stretching vibration peaks at $1021 \, cm^{-1}$ and $1241 \, cm^{-1}$ correspond to C–N and C–O, respectively. The deformation vibration peaks at $1435 \, cm^{-1}$ and $1638 \, cm^{-1}$ correspond to O–H and N–H, respectively, and the stretching vibration peaks at $1742 \, cm^{-1}$, $2929 \, cm^{-1}$ and $3406 \, cm^{-1}$ correspond to C=O, C–H and O–H, respectively. Compared with the plot of figure 4a, in the infrared spectrum of the modified PPP after adsorption of uranyl ions, the position of the absorption peak is shifted to a certain degree, and the intensity of the absorption peak is reduced. This phenomenon illustrates that the structure of AAUI has changed. The wavenumber of the O–H stretching vibration peak changed to $3428 \, cm^{-1}$, suggesting that its contribution to surface complexation reactions cannot be ignored [25]. The peak at $1742 \, cm^{-1}$ corresponds to C=O stretching vibration, which is obviously weakened, and this indicates that the influence of uranyl ions on the adsorbent of C=O was probably related to the complexation between them [26]. The characteristic peak is significantly enhanced and moving to high wavenumbers at near $1383 \, cm^{-1}$, and the peak corresponds to –CH symmetric bending vibrations –CHOH [27], suggesting that the complexation reaction occurs between –CHOH and $UO_2^{2+}$. The reason for the above change may be due to the adsorption of uranyl ions at the active site on the surface of PPP, replacing the $H^+$ on the surface group of the adsorbent.

### 3.6. Effect of pH

Batch adsorption tests were performed at pH ranging from 2 to 7. As shown in figure 5, these results show that pH has a significant effect on adsorption. When pH = 5.0, the adsorption efficiency and the

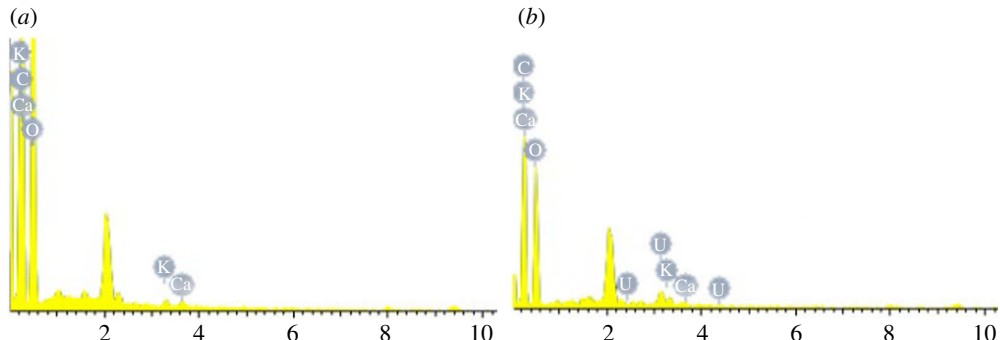

**Figure 2.** EDS spectra of chemical elements on modified PPP (*a*) before and (*b*) after adsorption for U(VI).

**Table 2.** Results of EDS for chemical elements on modified PPP before and after adsorption for U(VI).

| element | before | | after | |
| --- | --- | --- | --- | --- |
| | W % | At % | W % | At % |
| C K | 46.59 | 53.89 | 42.33 | 53.47 |
| O K | 52.89 | 45.93 | 48.29 | 45.80 |
| U M | 0 | 0 | 8.96 | 0.57 |
| K K | 0.28 | 0.10 | 0.05 | 0.02 |
| Ca K | 0.24 | 0.08 | 0.37 | 0.14 |

adsorption capacity of uranyl ions by the modified PPP adsorbent were the highest, 73.27% and 14.654 mg g$^{-1}$, respectively; increasing the pH from 2 to 5, the adsorption efficiency and the adsorption amount gradually increase; when pH = 5.0–6.0, the adsorption rate and the amount of adsorption change only slightly; when pH > 6.0, the adsorption rate and the amount of adsorption are lower. So the optimum pH for the adsorption of uranyl ions by modified PPP was 5.0–6.0.

Since the pH of the adsorption medium affects the ionization of the functional groups and the ion species, the influence of pH on adsorption is very large. When the solution is strongly acidic, H$^+$ in the solution has a positive charge and competes with uranyl ions for the active site of the adsorbent. H$^+$ first binds to the active site on the surface of the adsorbent, and uranyl ions are repelled and it is difficult to reach the active site, which leads to lower adsorption rates and capacities. When pH > 6.0, as the alkalinity of the solution increases, the hydroxide ion concentration in the solution continuously increases, and hydroxide ions and uranyl ions can form some ions with lower adsorption affinity, including $[UO_2OH]^+$, $[(UO_2)_3(OH)_5]^+$, $[(UO_2)_2(OH)_2(OH)_2]^{2+}$, $[(UO_2)_3(OH)_4]^{2+}$, $[(UO_2)_2OH]^{3+}$, $[(UO_2)_3(OH)]^{5+}$, $[(UO_2)_3(OH)_7]^-$ and $[UO_2(OH)_4]^{2-}$. The adsorption of uranyl ions decreases [7,28,29].

## 3.7. Effect of temperature

The effect of adsorption over the temperature range of 30–70°C was explored. As shown in figure 6, the result illustrates that the temperature has a significant influence on the adsorption. The adsorption rate and capacity increase with temperature. When the temperature is 70°C, the adsorption efficiency reaches 80.94% and the corresponding adsorption capacity is 16.188 mg g$^{-1}$. It can be seen that when the temperature is between 50°C and 70°C, the adsorption performance of $UO_2^{2+}$ is the best, and over this temperature range, the adsorption capacity and the adsorption rate change only slightly with the increase in temperature. For ease of operation, experiments were conducted at 50°C.

## 3.8. Effect of adsorbent dose

The effect of the mass of adsorbent dose on the adsorption was analysed. As shown in figure 7, by increasing the adsorbent dose from 250 to 1250 mg l$^{-1}$, the adsorption efficiency of uranyl ions gradually increased from 59.29% to 82.30%. The change in adsorption amount is opposite to the removal rate. With the increase in adsorbent dose, the adsorption amount decreases from 23.715 to

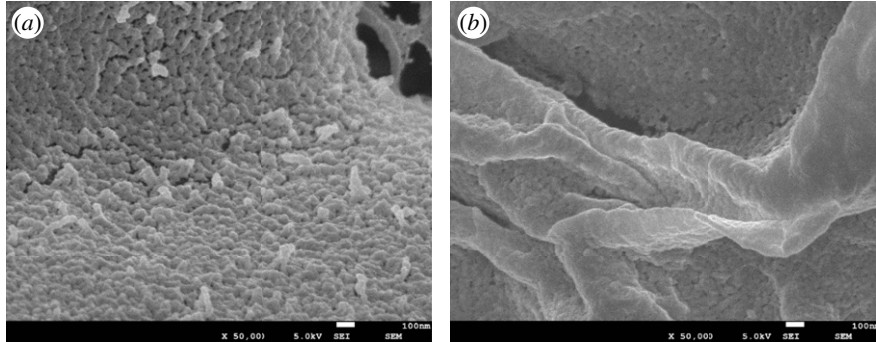

**Figure 3.** Microcosmic configuration of modified PPP (*a*) before and (*b*) after adsorption for U(VI).

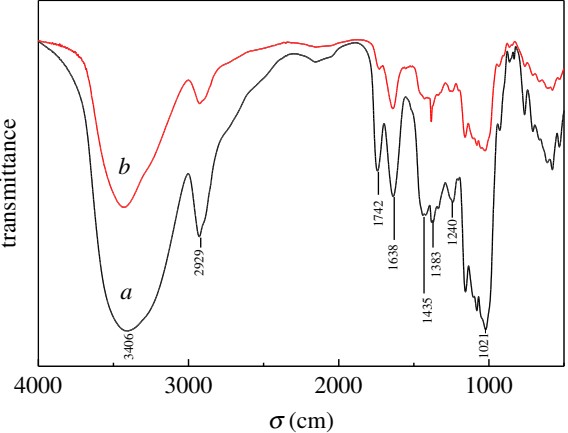

**Figure 4.** IR spectra of modified PPP (*a*) before and (*b*) after adsorption for U(VI).

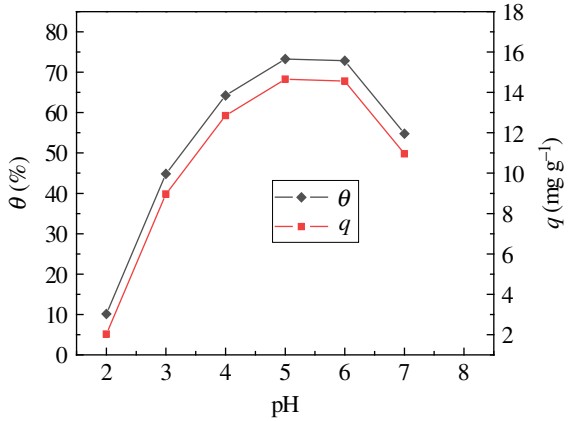

$C_0 = 10$ mg l$^{-1}$, $V = 20$ ml, $T = 50°C$, *m* = 500 mg l$^{-1}$, $t = 90$ min

**Figure 5.** Effect of pH on modified PPP adsorption of U(VI).

6.583 mg g$^{-1}$. The main reason for this is that with the amount of adsorbent increasing, the number of active sites for adsorption increases and the adsorption rate of uranyl ions can also gradually increase; however, the concentration of uranyl ions is constant, and with the mass of adsorbent increasing, the amount of adsorbed uranium per unit mass decreases, so that the amount of adsorption decreases.

## 3.9. Effect of contact time

The effect of different contact times (ranging from 15 to 150 min) was investigated. The results are shown in figure 8, indicating that the adsorption rate of uranyl ions in the modified PPP reached 71.92% when

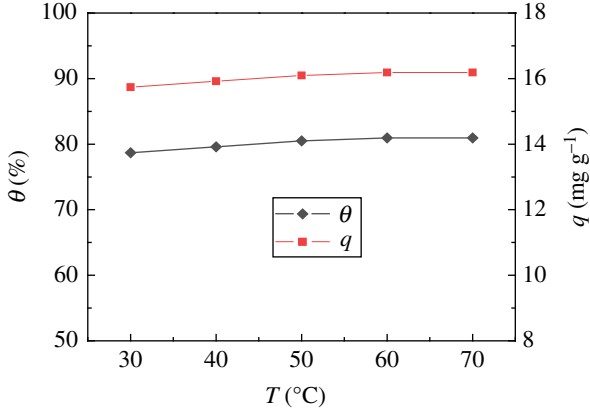

$C_0 = 10$ mg l$^{-1}$, V = 20 ml, *m = 500* mg l$^{-1}$, pH = 5.0, *t* = 90 min

**Figure 6.** Effect of temperature on modified PPP adsorption of U(VI).

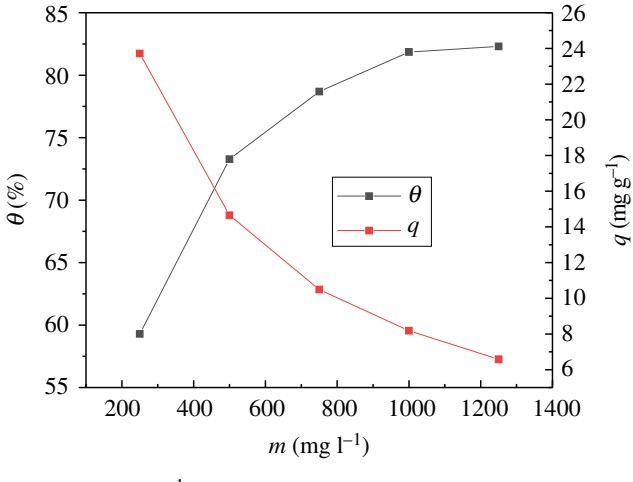

$C_0 = 10$ mg l$^{-1}$, V = 20 ml, *T* = 50°C, pH = 5.0, *t* = 90 min

**Figure 7.** Effect of adsorbent dose on adsorption of U(VI).

adsorption lasted for 30 min. The adsorption capacity and adsorption rates increase continuously until 60 min. In the flat section, the adsorption rate was 79.13%, and the adsorption capacity was 15.827 mg g$^{-1}$. At 90 min, the adsorption capacity and adsorption rate remained basically unchanged; the phenomenon confirms that the adsorption process has attained the adsorption equilibrium.

The adsorption of uranyl ions by modified PPP may take place on the outer surface and the inner surface of the sample [16,30]. The uranyl ions are first adsorbed on the surface of the sample. The adsorption efficiency is fast and the time required is short. However, uranyl ions need to be adsorbed on the inner surface, and two processes are required. First, the molecules diffuse into the cavities of the adsorbent, and then they are adsorbed. The adsorption of uranyl ions is relatively slow; therefore, the image tends to be gentle until adsorption reaches equilibrium.

## 3.10. Effect of initial concentration

The effect of the initial mass concentration of uranyl ions on the adsorption was analysed. The results are shown in figure 9, indicating that with the increase in the uranyl ion initial concentration, the adsorption rate of the modified PPP decreases, while the adsorption amount increases first and then decreases; when the initial concentration of uranyl ion reached 70 mg l$^{-1}$, the maximum adsorption capacity was 61.041 mg g$^{-1}$, but the adsorption rate was only 43.60%. When the initial mass concentration of uranyl ions was 5 mg l$^{-1}$, the adsorption rate was 76.31% and the adsorption capacity was only

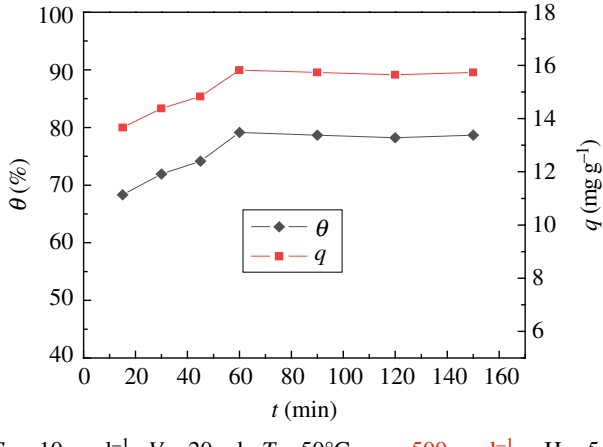

$C_0 = 10$ mg l$^{-1}$,  $V = 20$ ml,  $T = 50°C$,  $m = 500$ mg l$^{-1}$,  pH = 5.0

**Figure 8.** Effect of time on modified PPP adsorption of U(VI).

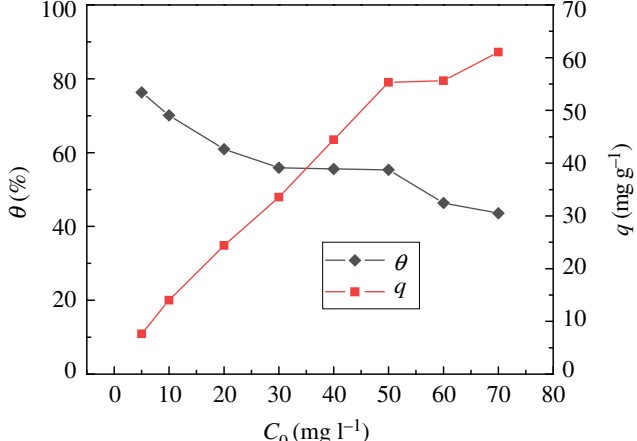

$V = 20$ ml,  $T = 50°C$,  $m = 500$ mg l$^{-1}$,  pH = 5.0,  $t = 90$ min

**Figure 9.** Effect of U(VI) initial concentration on modified PPP adsorption of U(VI).

7.631 mg g$^{-1}$. This phenomenon confirms that the initial mass concentration of uranyl ions has a significant influence on the adsorption performance of the modified PPP; PPP is adaptable to the treatment of medium and low concentrations of uranium-containing nuclear industrial wastewater.

When the initial concentration of uranyl ions in the solution is low, the uranyl ions can be in full contact with PPP, so the adsorption rate is higher; however, if the adsorbent's active site is not fully combined with uranyl ions, this results in low adsorption. As the initial concentration of uranyl ions continues to increase, the amount of adsorption increases significantly, and the adsorption rate decreases significantly. This is due to the increased concentration of uranyl ions, which is a gradual excess compared with the amount of adsorbent used. When the adsorption reaches saturation, the uranyl ion exists in the solution in a free state, resulting in a reduction in the adsorption rate, but the adsorption amount of uranium per unit mass becomes larger. Therefore, the amount of adsorption increases continuously.

## 3.11. Comparison of adsorption properties between modified and unmodified PPP

Through a series of adsorption experiments, the optimal conditions for the adsorption of uranyl ions by the modified PPP were obtained. Under this optimal condition, the uranyl ion adsorption experiments of modified and unmodified PPP were performed. As shown in figure 10, the results suggest that the adsorbed amount of the modified PPP can reach 42.733 mg g$^{-1}$, and the adsorption rate is 61.05%.

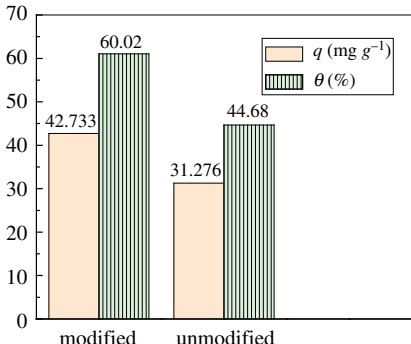

$C_0 = 35$ mg l$^{-1}$, $V = 20$ ml, $T = 50°C$, $m = 500$ mg l$^{-1}$, pH = 6.0, $t = 90$ min

**Figure 10.** Comparison of adsorption properties between modified and unmodified PPP.

**Table 3.** Kinetic parameters of U(VI) adsorption on modified PPP.

| model | $q$(mg g$^{-1}$) | $k$ | $R^2$ |
|---|---|---|---|
| pseudo first-order | 2.604 | 0.0026 (min$^{-1}$) | 0.8093 |
| pseudo second-order | 16.059 | 0.0232 (g mg$^{-1}$ min$^{-1}$) | 0.9995 |

The adsorption capacity of unmodified PPP and rate were only 31.276 mg g$^{-1}$ and 44.68%, respectively. The amount of adsorption increased by 26.8%, and the results confirm that the uranyl ion adsorption performance of the modified PPP significantly increased.

## 3.12. Kinetic parameters for adsorption of $UO_2^{2+}$

Adsorption kinetics can provide information about the adsorption mechanism. The adsorption rate can be fitted, the adsorption mechanism can be inferred, and a reasonable adsorption kinetics model can be established. The kinetic model is currently used for the pseudo first-order model (3.2) and the pseudo second-order model (3.3).

$$\lg(q_e - q_t) = \lg q_e - \frac{k_1}{2.303}t \tag{3.2}$$

and

$$\frac{t}{q_t} = \frac{1}{k_2 q_e^2} + \frac{t}{q_e}, \tag{3.3}$$

where $q_e$ is the equilibrium adsorption capacity (mg g$^{-1}$), $q_t$ is the capacity of adsorption at time $t$ (mg g$^{-1}$), $t$ is the adsorption time (min), $K_1$ is the pseudo first-order model adsorption rate constant (min$^{-1}$), and $K_2$ is the pseudo second-order model adsorption rate constant (g mg$^{-1}$ min$^{-1}$).

The above two adsorption kinetic models were used to fit the adsorption process. The kinetic parameters are given in table 3. As shown in figures 11 and 12, the pseudo second-order kinetics model ($R^2 = 0.9995$) is better than the pseudo first-order kinetics model ($R^2 = 0.8093$); the theoretical equilibrium adsorption capacity calculated by the pseudo second-order kinetics model is 16.059 mg g$^{-1}$. The theoretical equilibrium adsorption capacity calculated by the pseudo first-order kinetic model is 2.604 mg g$^{-1}$, and the equilibrium adsorption capacity obtained by the experiment was 15.827 mg g$^{-1}$. The results suggest that the theoretical adsorption capacity obtained by the pseudo second-order kinetic model is closer to the experimental value. Therefore, the adsorption process is more consistent with the pseudo second-order model, and confirms that the adsorption process is chemical adsorption [31,32].

## 3.13. Thermodynamic parameters for adsorption of $UO_2^{2+}$

Thermodynamic parameters, for instance, enthalpy change $\Delta H$ (kJ mol$^{-1}$), entropy change $\Delta S$ (J K$^{-1}$ mol$^{-1}$) and Gibbs free energy $\Delta G$ (kJ mol$^{-1}$), can be fitted by an adsorption thermodynamic

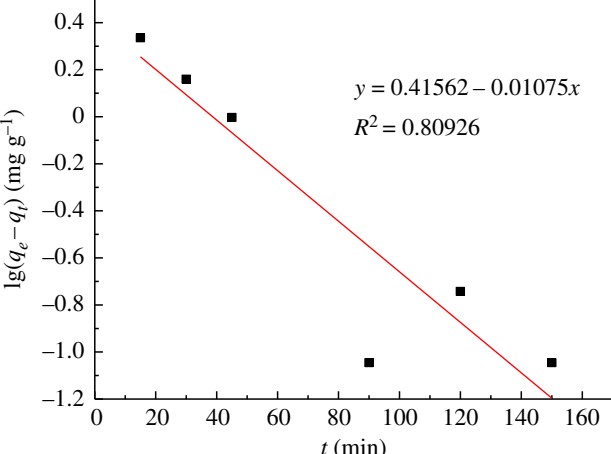

**Figure 11.** Pseudo first-order model.

$$y = 0.41562 - 0.01075x$$
$$R^2 = 0.80926$$

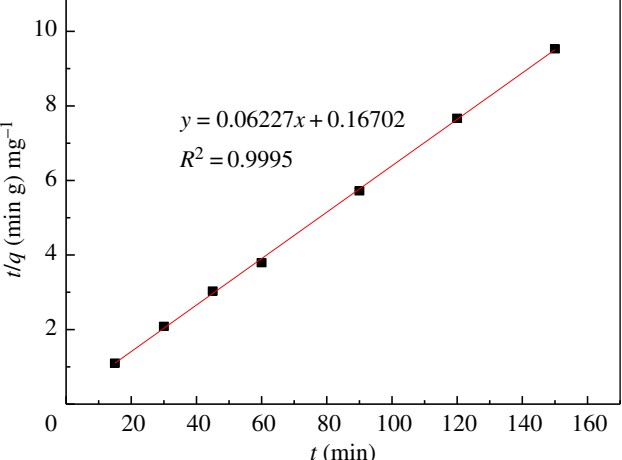

**Figure 12.** Pseudo second-order model.

$$y = 0.06227x + 0.16702$$
$$R^2 = 0.9995$$

model, which directly reflects the relationship between adsorbent and adsorbate molecules, and between adsorbent and solute. $\Delta H$, $\Delta S$ and $\Delta G$ are calculated by the following equations:

$$K_d = \frac{q_e}{C_e}, \tag{3.4}$$

$$\ln K_d = -\frac{\Delta H}{RT} + \frac{\Delta S}{R} \tag{3.5}$$

and
$$\Delta G = \Delta H - T\Delta S, \tag{3.6}$$

where $K_d$ is the adsorption partition coefficient (l mg$^{-1}$), $q_e$ is the equilibrium adsorption capacity (mg g$^{-1}$), $C_e$ is the equilibrium concentration (mg l$^{-1}$), $R$ is the ideal gas constant (8.314 J mol$^{-1}$ K$^{-1}$), $T$ is the temperature (K), $\Delta H$ is the enthalpy change (kJ mol$^{-1}$), $\Delta S$ is the entropy change (J K$^{-1}$ mol$^{-1}$), and $\Delta G$ is the Gibbs free energy (kJ mol$^{-1}$).

Next, $\ln K_d$ is mapped to $1/T$ and fitted to a straight line. The results are shown in figure 13. The slope and intercept were $\Delta H/R$ and $\Delta S/R$, respectively, and the thermodynamic parameters were obtained. The results are listed in table 4.

Table 4 testifies that the adsorption process $\Delta H > 0$, confirming that the adsorption process is an endothermic process; $\Delta S > 0$, indicating that the degree of freedom of the process is increased, and the degree of dislocation between the solid and liquid phases in the adsorption process is increased, which may be due to the formation of stable complexes on the surface of PPP when uranyl ions were adsorbed on the PPP. The $\Delta G$ values at the three temperatures shown in the table are negative, showing that the adsorption process is spontaneous, and as the temperature increases, the absolute value of $\Delta G$ increases, that is, as the temperature increases the spontaneous degree of the adsorption

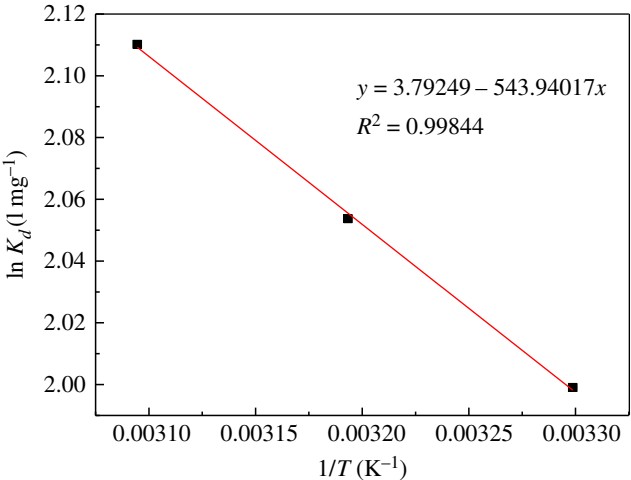

$$y = 3.79249 - 543.94017x$$
$$R^2 = 0.99844$$

**Figure 13.** Plot of $\ln K_d$ versus 1/T.

**Table 4.** Thermodynamic parameters of U(VI) adsorption on modified PPP.

| $\Delta H$ (kJ mol$^{-1}$) | $\Delta S$ (J k$^{-1}$ mol$^{-1}$) | $\Delta G$ (kJ mol$^{-1}$) | | |
|---|---|---|---|---|
| | | 303.15 K | 313.15 K | 323.15 K |
| 4.522 | 31.531 | $-5.036$ | $-5.352$ | $-5.667$ |

**Table 5.** Langmuir and Freundlich parameters of U(VI) adsorption on modified PPP.

| Langmuir | | | Freundlich | | |
|---|---|---|---|---|---|
| $q_0$ | $K_L$ | $R^2$ | $n$ | $K_F$ | $R^2$ |
| 84.388 | 0.064 | 0.946 | 1.6778 | 7.134 | 0.994 |

process increases. The data in the table also show that as at all temperatures $\Delta G$ is a negative value, the adsorption process is an entropy-dominated rather than a helium-dominated process [33,34].

## 3.14. Adsorption isotherm

Furthermore, the adsorption process was investigated; the Langmuir isotherm equation and the Freundlich isotherm equation were used for the fitting.

Langmuir isotherm equation:

$$\frac{C_e}{q_e} = \frac{1}{K_L q_0} + \frac{C_e}{q_0}, \tag{3.7}$$

Freundlich isotherm equation:

$$\lg q_e = \lg K_F + \frac{1}{n}\lg C_e, \tag{3.8}$$

where $C_e$ is the equilibrium concentration of uranyl ions (mg l$^{-1}$), $q_e$ is the equilibrium adsorption capacity (mg g$^{-1}$), $q_0$ is the maximum adsorption capacity (mg g$^{-1}$), $K_L$ is the Langmuir adsorption equilibrium constant, $K_F$ is the Freundlich adsorption coefficient, and $n$ is the Freundlich constant.

Two isothermal adsorption equations were used for fitting. The parameters are given in table 5. These results are shown in figures 14 and 15.

From the data shown, the Freundlich adsorption model ($R^2 = 0.994$) has a higher degree of fit than the Langmuir model ($R^2 = 0.946$). The characteristic parameters $n > 1$, $1/n = 0.599$, indicating that the modified PPP has good uranyl ion adsorption performance; the Langmuir isotherm model provides two

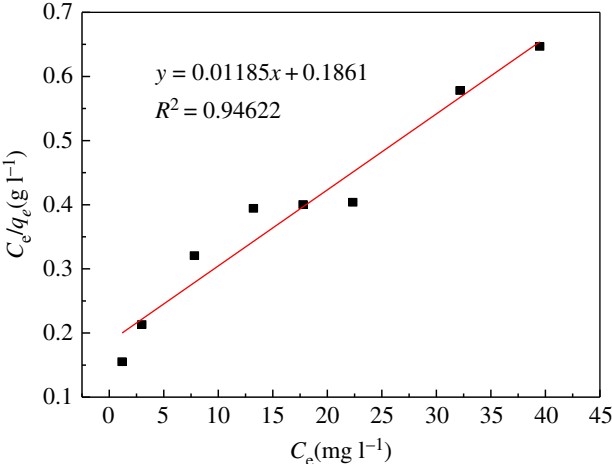

**Figure 14.** Langmuir adsorption isotherm.

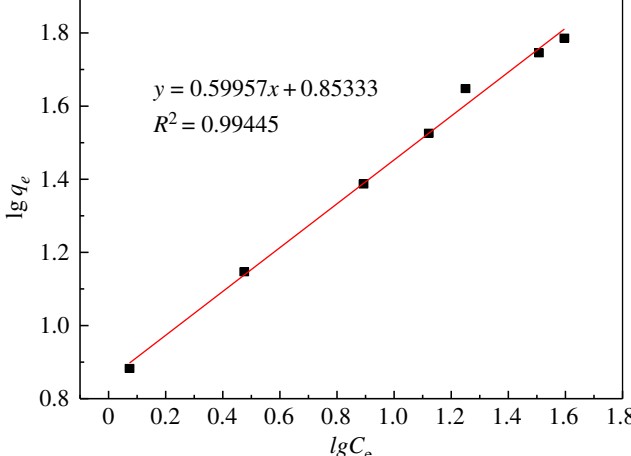

**Figure 15.** Freundlich adsorption isotherm.

parameters, the maximum adsorption capacity ($q_0$) and Langmuir adsorption equilibrium. The constant ($K_L$), by fitting, shows that the maximum adsorption capacity which can be reached is 84.388 mg g$^{-1}$. In summary, the adsorption process is in good agreement with the Freundlich adsorption model [35].

## 3.15. Desorption and reusability of the modified PPP

The above adsorption experiment steps were repeated to perform a cycle regeneration experiment: 100 ml of 0.50 mol l$^{-1}$ NaOH was designated as an eluent for desorption of UO$_2^{2+}$ from modified PPP. Figure 16 indicates that the adsorption capacity of the modified PPP decreased from 42.733 mg g$^{-1}$ in the first cycle to 40.121 mg g$^{-1}$ in the fifth cycle. After five cycles of adsorption and regeneration, the performance of the modified PPP did not decrease significantly. The above data suggest that the modified PPP is a kind of repeatable adsorption material after regeneration.

## 3.16. Mechanisms

The kinetic and thermodynamic experiments show that the adsorption process is in good agreement with the pseudo second-order kinetics model and it is an endothermic reaction, and the reaction is spontaneous, confirming that the adsorption process is chemical adsorption. In addition, the adsorption mechanism of modified PPP for uranyl ions was further proved by the analysis of IR. The results indicate that the modified PPP may achieve the purpose of removing UO$_2^{2+}$ from solutions through surface complexation and ion exchange [27,30].

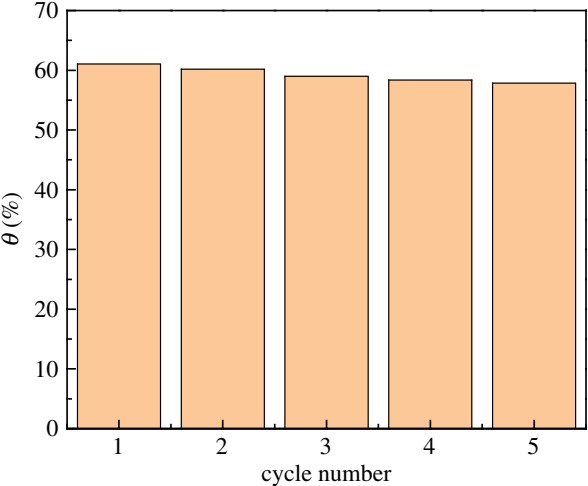

**Figure 16.** Effect of cycle times on adsorption performance.

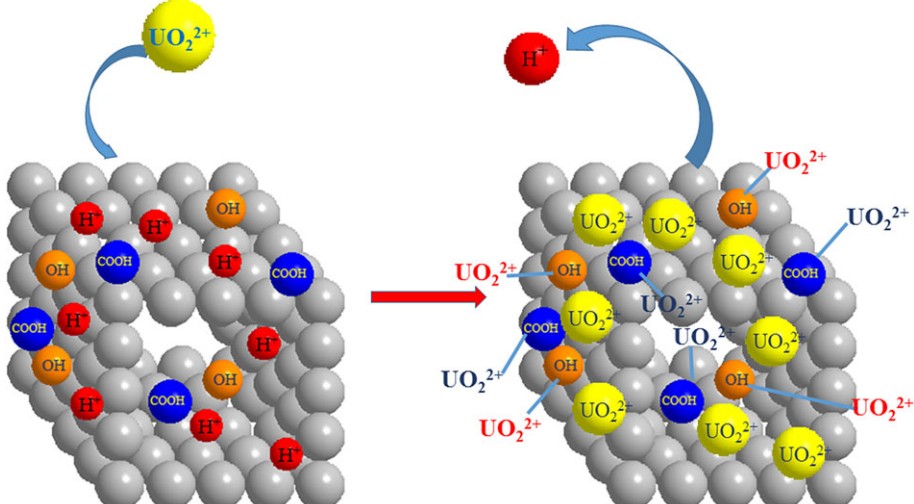

**Figure 17.** The adsorption mechanism of modified PPP for uranylions.

Therefore, the sorption mechanism can be expressed by the following reactions:

$$R-COOH + UO_2^2+ \rightarrow R-COOH~(UO_2)^{2+}, \tag{3.9}$$

$$R-CHOH + UO_2^2+ \rightarrow R-CHOH~(UO_2)^{2+} \tag{3.10}$$

$$\text{and} \quad R-H^+ + UO_2^2+ \rightarrow R-UO_2^{2+} + H^+. \tag{3.11}$$

The possible mechanisms of the adsorption process are shown in figure 17.

# 4. Conclusion

The modified PPP and AAUI have been investigated by BET, SEM, IR and EDS. The factors of different conditions such as pH value, temperature, mass of sorbent, adsorption time, and initial concentration of $UO_2^{2+}$ on the adsorption of uranyl ions were studied. The results of adsorption thermodynamics and adsorption kinetics experiments were as follows:

(1) SEM showed that the surface of the modified PPP had good porous structure, rough surface, continuous and uniform porous structure, which was conducive to adsorption;

(2) The optimum pH for the adsorption of uranyl ions by modified PPP was 5.0–6.0; when the initial mass concentration of $UO_2^{2+}$ was 35 mg l$^{-1}$ and the pH value was 6.0, the dosage of adsorbent

was 500 mg l$^{-1}$, temperature 50°C and adsorption time 90 min, the adsorbed capacity of uranyl ions by the modified PPP adsorbent can reach 42.733 mg g$^{-1}$, the adsorption rate is 61.05%, while the adsorption capacity of unmodified adsorbent was only 31.276 mg g$^{-1}$, and the adsorption rate was 44.68%. The amount of adsorption increased by 26.8%, thus the results confirm that the uranyl ion adsorption performance of the modified PPP is obviously enhanced.

(3) The kinetic and thermodynamic experiments show that the adsorption process is in good agreement with the pseudo second-order kinetics model and it is an endothermic reaction, and the reaction is spontaneous. The adsorption process is entropy-dominated. The Freundlich adsorption isotherm can describe the adsorption process more accurately. The primary mechanism of adsorption was surface complexation and ion exchange.

Data accessibility. The datasets supporting this article have been uploaded as part of the electronic supplementary material (http://dx.doi.org/10.5061/dryad.rg27v0h).

Authors' contributions. Conceived and designed the experiments: P.Y. and Y.X.. Performed the experiments: J.T., A.L. and L.L. Analysed the data: Y.X., A.L., L.L. and H.S. Contributed reagents/materials/analysis tools: P.Y., Y.X. and J.T. Wrote the paper: P.Y., Y.X. and J.T. Copyedited the manuscript: P.Y. and Y.X.

Competing interests. We have no competing interests.

Funding. This study was financially supported by the Natural Science Foundation of Hunan Province (2017JJ2231) and Hunan Province Engineering Research Center of Radioactive Control Technology in Uranium Mining and Metallurgy, and Hunan Province Engineering Technology Research Center of Uranium Tailings Treatment Technology, University of South China (2019YKZX1007).

Acknowledgements. We gratefully acknowledge the anonymous reviewers for their constructive comments which have improved the quality of this paper. We also acknowledge Dr Liu Yong (University of South China) for his assistance.

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
