## [Reviewer comments · Royal Society Open Science]

Review History

RSOS-181986.R0 (Original submission)

Review form: Reviewer 1

Is the manuscript scientifically sound in its present form?

Yes

Are the interpretations and conclusions justified by the results?

Yes

Is the language acceptable?

Yes

Is it clear how to access all supporting data?

Yes

Do you have any ethical concerns with this paper?

No

Have you any concerns about statistical analyses in this paper?

No

Recommendation?

Major revision is needed (please make suggestions in comments)

Comments to the Author(s)

In this manuscript, the authors modified PPP and characterized the PPP using different techniques in detail. The prepared PPP was applied as adsorbent for the removal of U(VI) from aqueous solutions under different conditions and discussed the results in detail. The contents are important for the removal of U(VI) from aqueous solutions. After reading the manuscript, I think it can be accepted for publication after revision.

Special comments:

1. For all the references, please add the journal names to each references. Also please edit the names of the authors in the references.
2. Please separate the authors in author list.
3. In the Introduction, several critical reviews for the removal of U(VI) from aqueous solutions should be added in the revised form as this is important for readers in this area such as Chemical Society Reviews. 2018, 47, 2322-2356; Environmental Pollution. 2018, 240, 493-505; Polymer Chemistry. 2018, 9, 3562-3582.
4. In the Summary section, the dosage of adsorbent (10mg) is not correct, it should be 10mg/L or 10 mg/mL.... The volume of the solution should be given.
5. Did the authors check the precipitation of U(VI) at pH 6.0?
6. What is the main mechanism of the sorption of U(VI)? Several important papers are helpful to improve the discussion of the results such as Chemical Engineering Journal. 2019, 355, 697-709; 2019, 359, 1550-1562; 2018, 332, 775-786; 2018, 338, 579-590; 2018, 346, 406-415; 2018, 343, 460-466; 2018, 353, 360-370.

Review form: Reviewer 2 (Osman Uner)

Is the manuscript scientifically sound in its present form?

Yes

Are the interpretations and conclusions justified by the results?

Yes

Is the language acceptable?

Yes

Is it clear how to access all supporting data?

Yes

Do you have any ethical concerns with this paper?

No

Have you any concerns about statistical analyses in this paper?

No

Recommendation?

Major revision is needed (please make suggestions in comments)

Comments to the Author(s)

In this manuscript, the authors prepared a chemically modified PPP by using Pomelo peel's pulp (PPP). The adsorbent has been characterized by using EDS, IR, BET and SEM. Also, UO₂²⁺ adsorption by the modified PPP was studied at different conditions such as pH, mass of adsorbent, adsorption time, initial concentration, temperature, and so on. Moreover, its kinetic and thermodynamic parameters were investigated. At the first reading the manuscript, it sounds scientifically good to me, but there are various missing explanations and mistakes after a few readings. Therefore, major revision is needed before the publication at Royal Society Open Science. The authors must consider the following comments.

1- In lines between 17 and 27 on page 4, in the section of 2.Introduction, "Tao Dong et al. [18] used barium metaphosphate as an adsorbent to study the adsorption properties of uranyl ions under various conditions and the maximum adsorption capacity can reach 261.78 mg/g under the optimal conditions; Zhang Zongbo et al [19] synthesized polyamino-containing ligands modified ordered mesoporous materials with surface area 1052 cm²/g, and the adsorption of uranium can reach 454 mg/g. Although the above studies obtain excellent results on adsorption of uranyl ions, they have disadvantages about high costs and secondary pollution."

You determined the adsorbed amount of uranyl ion by the modified PPP adsorbent as 42.733 mg/g. Tao Dong et al. found 6 times higher uranyl ion adsorption capacity, and Zhang Zongbo et al found higher than 10 times. You claim that these studies have disadvantages about high costs and secondary pollution. Did you calculate their costs and also your cost? You used various materials and chemicals to obtain the modified PPP. Moreover, experiments were conducted at 50 °C. Also, what is secondary pollution that these studies have but your study does not have.

2- In lines between 11 and 15 on page 5, in the section of 4.2. BET, "According to the data in the table, the BET surface area of the modified PPP is 36.733 m²/g, indicating the modified PPP adsorbent with a large BET surface area which has potential adsorption properties for UO₂²⁺." How do you know that indicating the modified PPP adsorbent with a large BET surface area which has potential adsorption properties for UO₂²⁺ only from Table 1? Also, is a BET surface area of 36.733 m²/g high for you? What is your criterion to be a high BET surface area?

3- After line 60 on page 5, in the section of 4.5. IR, "... a distinct O–U characteristic peak appeared at the wavenumber of 1383 cm⁻¹." The peak at 1383 cm⁻¹ is already present at the infrared spectrum of the modified PPP in Figure 4(a). You should give more clear explanations for all the peaks, and you should cite all the peaks that were given in previous studies.

4- In lines between 27 and 32 on page 8, in the section of 4.6. Effect of pH, "When pH>5.0, as the alkalinity of the solution increases, the hydroxide ion concentration in the solution continuously increases, hydroxide ion and uranyl ions can form some ions with lower adsorption affinity ..." However, in Fig.10 whose caption is comparison of adsorption properties between modified and unmodified PPP, the parameters are C₀=35 mg/L, V=20 mL, T=50°C, m=10 mg, pH=6.0, t=90 min. Why did you adjust pH to 6.0? Also, readers would want to see uranyl ion adsorption by the modified PPP at pH>7.0.

5- There are many papers related to the adsorption of uranium ions. Therefore, authors can add the comparisons of uranium ion adsorption capacities by different adsorbents. Also, their costs and efficiencies can be added.

Review form: Reviewer 3

Is the manuscript scientifically sound in its present form?

Yes

Are the interpretations and conclusions justified by the results?

Yes

Is the language acceptable?

Yes

Is it clear how to access all supporting data?

Yes

Do you have any ethical concerns with this paper?

No

Have you any concerns about statistical analyses in this paper?

No

Recommendation?

Major revision is needed (please make suggestions in comments)

Comments to the Author(s)

The authors used the PPP as raw materials to prepare the adsorbent. The factors of different adsorption conditions such as pH value, adsorption temperature, mass of adsorption time, and initial concentration of UO_2^{2+} were investigated. This work has a certain innovation. However, there are several imperfections in the manuscript and the experimental results haven't presented well. Therefore, a major revision should be finished in order to suitable for these journals, and the detailed comments are as follows:

1. The introduction parts should re-write.
2. Page 7, line 10, the specific surface area, average pore diameters, and pore volume of the unmodified PPP should be added in the Table 1.
3. Page 8, line 46, it can be seen that when the temperature is between 50-70 °C, the adsorption performance of UO_2^{2+} is the best. Typically, in adsorption experiment, we may like to choose room temperature. The author said the best temperature is 50-70 °C. Please give the reasons.
4. Page 9, line 3, with the increase of adsorbent dose, the adsorption amount decrease from 23.715 mg/g to 6.583 mg/g. Please give lots of experimental data to prove it.
5. Page 9, line 29, the adsorption of uranyl ions by modified PPP mainly takes place on the outer surface and the inner surface of the sample. Please give the reasons.
6. Page 19, fig. 6 should be improved.
7. The authors should prove the recyclability of the absorbed by performing recycling experiments
8. There exist some minor errors including spelling grammar and format for this paper.

Review form: Reviewer 4

Is the manuscript scientifically sound in its present form?

No

Are the interpretations and conclusions justified by the results?

Yes

Is the language acceptable?

Yes

Is it clear how to access all supporting data?

Not Applicable

Do you have any ethical concerns with this paper?

No

Have you any concerns about statistical analyses in this paper?

No

Recommendation?

Reject

Comments to the Author(s)

The authors made a new chemical modified PPP from pomelo pulp (PPP) by fermentation, cooking and freeze-drying. The adsorbent was characterized by means of SEM, IR, BET and EDS. The effects of different adsorption conditions (pH, adsorption temperature, adsorbent quality, adsorption time, initial concentration of UO₂ 2+) on the adsorbent performance were studied. The experiment of this paper is feasible, but there is no obvious innovation. There are many results and papers about similar modified adsorbents. The adsorption mechanism was discussed but not enough. The optimum adsorption conditions obtained by the author are relatively common, and there is no new knowledge about adsorption behavior. However, the PPP chosen by the authors, as a waste, can be used for other purposes rather than simply adsorption. I suggest that they have an in-depth understanding of the potential of this material and make full use of it in a more appropriate place (or valuable products, such as chemicals) instead of using complex steps to make water treatment adsorbents.

Decision letter (RSOS-181986.R0)

14-Jan-2019

Dear Dr Pengfei:

Title: Preparation of Modified Pomelo Peel's Pulp Adsorbent and Its Adsorption to Uranyl Ions
Manuscript ID: RSOS-181986

The editor assigned to your manuscript has now received comments from reviewers. We would like you to revise your paper in accordance with the referee and Subject Editor suggestions which can be found below (not including confidential reports to the Editor). Please note this decision does not guarantee eventual acceptance.

Please submit your revised paper before 06-Feb-2019. Please note that the revision deadline will expire at 00.00am on this date. If we do not hear from you within this time then it will be assumed that the paper has been withdrawn. In exceptional circumstances, extensions may be possible if agreed with the Editorial Office in advance. We do not allow multiple rounds of revision so we urge you to make every effort to fully address all of the comments at this stage. If deemed necessary by the Editors, your manuscript will be sent back to one or more of the original reviewers for assessment. If the original reviewers are not available we may invite new reviewers.

To revise your manuscript, log into <http://mc.manuscriptcentral.com/rsos> and enter your Author Centre, where you will find your manuscript title listed under "Manuscripts with Decisions." Under "Actions," click on "Create a Revision." Your manuscript number has been

appended to denote a revision. Revise your manuscript and upload a new version through your Author Centre.

Please also include the following statements alongside the other end statements. As we cannot publish your manuscript without these end statements included, if you feel that a given heading is not relevant to your paper, please nevertheless include the heading and explicitly state that it is not relevant to your work.

- Ethics statement

Please clarify whether you received ethical approval from a local ethics committee to carry out your study. If so please include details of this, including the name of the committee that gave consent in a Research Ethics section after your main text. Please also clarify whether you received informed consent for the participants to participate in the study and state this in your Research Ethics section.

OR

Please clarify whether you obtained the necessary licences and approvals from your institutional animal ethics committee before conducting your research. Please provide details of these licences and approvals in an Animal Ethics section after your main text.

OR

Please clarify whether you obtained the appropriate permissions and licences to conduct the fieldwork detailed in your study. Please provide details of these in your methods section.

- Acknowledgements

RSC Associate Editor:
Comments to the Author:

(There are no comments.)

RSC Subject Editor:

Comments to the Author:

(There are no comments.)

Reviewers' Comments to Author:

Reviewer: 1

Comments to the Author(s)

In this manuscript, the authors modified PPP and characterized the PPP using different techniques in detail. The prepared PPP was applied as adsorbent for the removal of U(VI) from aqueous solutions under different conditions and discussed the results in detail. The contents are important for the removal of U(VI) from aqueous solutions. After reading the manuscript, I think it can be accepted for publication after revision.

Special comments:

1. For all the references, please add the journal names to each references. Also please edit the names of the authors in the references.
2. Please separate the authors in author list.
3. In the Introduction, several critical reviews for the removal of U(VI) from aqueous solutions should be added in the revised form as this is important for readers in this area such as Chemical Society Reviews. 2018, 47, 2322-2356; Environmental Pollution. 2018, 240, 493-505; Polymer Chemistry. 2018, 9, 3562-3582.
4. In the Summary section, the dosage of adsorbent (10mg) is not correct, it should be 10mg/L or 10 mg/mL.... The volume of the solution should be given.
5. Did the authors check the precipitation of U(VI) at pH 6.0?
6. What is the main mechanism of the sorption of U(VI)? Several important papers are helpful to improve the discussion of the results such as Chemical Engineering Journal. 2019, 355, 697-709; 2019, 359, 1550-1562; 2018, 332, 775-786; 2018, 338, 579-590; 2018, 346, 406-415; 2018, 343, 460-466; 2018, 353, 360-370.

Reviewer: 2

Comments to the Author(s)

In this manuscript, the authors prepared a chemically modified PPP by using Pomelo peel's pulp (PPP). The adsorbent has been characterized by using EDS, IR, BET and SEM. Also, UO₂²⁺ adsorption by the modified PPP was studied at different conditions such as pH, mass of adsorbent, adsorption time, initial concentration, temperature, and so on. Moreover, its kinetic and thermodynamic parameters were investigated. At the first reading the manuscript, it sounds scientifically good to me, but there are various missing explanations and mistakes after a few readings. Therefore, major revision is needed before the publication at Royal Society Open Science. The authors must consider the following comments.

- 1- In lines between 17 and 27 on page 4, in the section of 2.Introduction, "Tao Dong et al. [18] used barium metaphosphate as an adsorbent to study the adsorption properties of uranyl ions under various conditions and the maximum adsorption capacity can reach 261.78 mg/g under the optimal conditions; Zhang Zongbo et al [19] synthesized polyamino-containing ligands modified ordered mesoporous materials with surface area 1052 cm²/g, and the adsorption of uranium can reach 454 mg/g. Although the above studies obtain excellent results on adsorption of uranyl ions, they have disadvantages about high costs and secondary pollution."

You determined the adsorbed amount of uranyl ion by the modified PPP adsorbent as 42.733 mg/g. Tao Dong et al. found 6 times higher uranyl ion adsorption capacity, and Zhang Zongbo et al found higher than 10 times. You claim that these studies have disadvantages about high costs and secondary pollution. Did you calculate their costs and also your cost? You used various materials and chemicals to obtain the modified PPP. Moreover, experiments were conducted at 50 °C. Also, what is secondary pollution that these studies have but your study does not have.

2- In lines between 11 and 15 on page 5, in the section of 4.2. BET, "According to the data in the table, the BET surface area of the modified PPP is 36.733 m²/g, indicating the modified PPP adsorbent with a large BET surface area which has potential adsorption properties for UO₂²⁺." How do you know that indicating the modified PPP adsorbent with a large BET surface area which has potential adsorption properties for UO₂²⁺ only from Table 1? Also, is a BET surface area of 36.733 m²/g high for you? What is your criterion to be a high BET surface area?

3- After line 60 on page 5, in the section of 4.5. IR, "... a distinct O–U characteristic peak appeared at the wavenumber of 1383 cm⁻¹." The peak at 1383 cm⁻¹ is already present at the infrared spectrum of the modified PPP in Figure 4(a). You should give more clear explanations for all the peaks, and you should cite all the peaks that were given in previous studies.

4- In lines between 27 and 32 on page 8, in the section of 4.6. Effect of pH, "When pH>5.0, as the alkalinity of the solution increases, the hydroxide ion concentration in the solution continuously increases, hydroxide ion and uranyl ions can form some ions with lower adsorption affinity ...". However, in Fig.10 whose caption is comparison of adsorption properties between modified and unmodified PPP, the parameters are C₀=35 mg/L, V=20 mL, T=50°C, m=10 mg, pH=6.0, t=90 min. Why did you adjust pH to 6.0? Also, readers would want to see uranyl ion adsorption by the modified PPP at pH>7.0.

5- There are many papers related to the adsorption of uranium ions. Therefore, authors can add the comparisons of uranium ion adsorption capacities by different adsorbents. Also, their costs and efficiencies can be added.

Reviewer: 3

Comments to the Author(s)

The authors used the PPP as raw materials to prepare the adsorbent. The factors of different adsorption conditions such as pH value, adsorption temperature, mass of adsorption time, and initial concentration of UO₂²⁺ were investigated. This work has a certain innovation. However, there are several imperfections in the manuscript and the experimental results haven't presented well. Therefore, a major revision should be finished in order to suitable for these journals, and the detailed comments are as follows:

1. The introduction parts should re-write.
2. Page 7, line 10, the specific surface area, average pore diameters, and pore volume of the unmodified PPP should be added in the Table 1.
3. Page 8, line 46, it can be seen that when the temperature is between 50-70 °C, the adsorption performance of UO₂²⁺ is the best. Typically, in adsorption experiment, we may like to choose room temperature. The author said the best temperature is 50-70 °C. Please give the reasons.
4. Page 9, line 3, with the increase of adsorbent dose, the adsorption amount decrease from 23.715 mg/g to 6.583 mg/g. Please give lots of experimental data to prove it.
5. Page 9, line 29, the adsorption of uranyl ions by modified PPP mainly takes place on the outer surface and the inner surface of the sample. Please give the reasons.
6. Page 19, fig. 6 should be improved.
7. The authors should prove the recyclability of the absorbed by performing recycling experiments
8. There exist some minor errors including spelling grammar and format for this paper.

Reviewer: 4

Comments to the Author(s)

The authors made a new chemical modified PPP from pomelo pulp (PPP) by fermentation, cooking and freeze-drying. The adsorbent was characterized by means of SEM, IR, BET and EDS. The effects of different adsorption conditions (pH, adsorption temperature, adsorbent quality, adsorption time, initial concentration of UO_2^{2+}) on the adsorbent performance were studied. The experiment of this paper is feasible, but there is no obvious innovation. There are many results and papers about similar modified adsorbents. The adsorption mechanism was discussed but not enough. The optimum adsorption conditions obtained by the author are relatively common, and there is no new knowledge about adsorption behavior. However, the PPP chosen by the authors, as a waste, can be used for other purposes rather than simply adsorption. I suggest that they have an in-depth understanding of the potential of this material and make full use of it in a more appropriate place (or valuable products, such as chemicals) instead of using complex steps to make water treatment adsorbents.

Author's Response to Decision Letter for (RSOS-181986.R0)

See Appendix A.

Decision letter (RSOS-181986.R1)

08-Feb-2019

Dear Dr pengfei:

Title: Preparation of Modified Pomelo Peel's Pulp Adsorbent and Its Adsorption to Uranyl Ions
Manuscript ID: RSOS-181986.R1

It is a pleasure to accept your manuscript in its current form for publication in Royal Society Open Science. The chemistry content of Royal Society Open Science is published in collaboration with the Royal Society of Chemistry.

Royal Society of Chemistry
Thomas Graham House
Science Park, Milton Road
Cambridge, CB4 0WF

Royal Society Open Science - Chemistry Editorial Office

RSC Associate Editor
Comments to the Author:
(There are no comments.)

Reviewer(s)' Comments to Author:

Appendix A

Response to Referees

Dear Editors and Reviewers:

Thank you for your letter and for the reviewers' comments concerning our manuscript entitled "Preparation of Modified Pomelo Peel's Pulp Adsorbent and Its Adsorption to Uranyl Ions". Those comments are all valuable and very helpful for revising and improving our manuscript, as well as the important guiding significance to our researches. We have studied comments carefully and have made correction which we hope meet with approval. Revised portion are marked in red in the manuscript. The main corrections and the responds to the reviewer's comments are as following:

Responds to the reviewer's comments:

Reviewer #1:

1. For all the references, please add the journal names to each references. Also please edit the names of the authors in the references.

Response: Thank you for reading carefully. We have added the information and marked them red in the revised manuscript.

2. Please separate the authors in author list.

Response: Thank you for your suggestion. We have corrected the error and marked them red in the revised manuscript.

3. In the Introduction, several critical reviews for the removal of U(VI) from aqueous solutions should be added in the revised form as this is important for readers in this area such as Chemical Society Reviews. 2018, 4^[1,2]7, 2322-2356; Environmental Pollution. 2018, 240, 493-505; Polymer Chemistry. 2018, 9, 3562-3582.

Response: Thank you for your suggestion. We have added these critical reviews and marked them red in the revised manuscript.

4. In the Summary section, the dosage of adsorbent (10mg) is not correct, it should be 10mg/L or 10 mg/mL.... The volume of the solution should be given.

Response: Thank you to give the proper word, We have corrected the error and marked them red in the revised manuscript.

5. Did the authors check the precipitation of U(VI) at pH 6.0?

Response: Thank you for your careful reading. In our experiment, when the pH of the solution was 6.0, we did not find the precipitation of U(VI) in the solution we prepared.

6. What is the main mechanism of the sorption of U(VI)?

Response: Thank you for your suggestion. The kinetic and thermodynamic experiments show that the adsorption processes is in good agreement to the pseudo-second-order kinetics model and is an endothermic reaction, and the reaction is spontaneous, confirm that the adsorption process is chemical adsorption. In addition, The adsorption mechanism of modified PPP for uranyl ions was further proved by the analysis of IR. The results indicate that the modified PPP may achieve the purpose of removing UO_2^{2+} from solutions through surface complexation and ion exchange. In the article, we add relevant explanations.

Reviewer: 2

1. In lines between 17 and 27 on page 4, in the section of 2. Introduction, “Tao Dong et al. [18] used barium metaphosphate as an adsorbent to study the adsorption properties of uranyl ions under various conditions and the maximum adsorption capacity can reach 261.78 mg/g under the optimal conditions; Zhang Zongbo et al [19] synthesized polyamino-containing ligands modified ordered mesoporous materials with surface area $1052 \text{ cm}^2/\text{g}$, and the adsorption of uranium can reach 454 mg/g. Although the above studies obtain excellent results on adsorption of uranyl ions, they have disadvantages about high costs and secondary pollution.” You determined the adsorbed amount of uranyl ion by the modified PPP adsorbent as 42.733 mg/g. Tao Dong et al. found 6 times higher uranyl ion adsorption capacity, and Zhang Zongbo et al found

higher than 10 times. You claim that these studies have disadvantages about high costs and secondary pollution. Did you calculate their costs and also your cost? You used various materials and chemicals to obtain the modified PPP. Moreover, experiments were conducted at 50 °C. Also, what is secondary pollution that these studies have but your study does not have.

Response: Thank you for your suggestion. After our investigation, we found that most of their research is synthetic materials, raw materials are not easy to obtain, and the raw materials of ppp materials we prepared are cheap and easy to get Pomelo Peel. It is well known that the production of Pomelo is extremely rich in China. Simultaneously, this also produces a large amount of agricultural waste. It can be said that the raw material of Pomelo peel as adsorbent is relatively low cost, and it also reduces environmental pollution. Hence, our research aims to make full use of large amounts of agricultural waste and provide a new material for the treatment of nuclear industry wastewater. Some of the less rigorous words have been corrected and marked them red in the revised manuscript.

2. In lines between 11 and 15 on page 5, in the section of 4.2. BET, “According to the data in the table, the BET surface area of the modified PPP is 36.733 m²/g, indicating the modified PPP adsorbent with a large BET surface area which has potential adsorption properties for UO₂²⁺.” How do you know that indicating the modified PPP adsorbent with a large BET surface area which has potential adsorption properties for

UO₂²⁺ only from Table 1? Also, is a BET surface area of 36.733 m²/g high for you? What is your criterion to be a high BET surface area?

Response: Thank you for your valuable and thoughtful comments. We are sorry to have an unclear writing. we have rewritten this part and marked red in the revised manuscript.

It is well known that the larger the specific surface area of the adsorbent, so the surface active sites will increase, simultaneously, this is advantageous for adsorption. The PPP becomes fluffy by chemical modification, and the specific surface area also increases, so we think that the modified PPP has potential adsorption properties for uranyl ions.

3. After line 60 on page 5, in the section of 4.5. IR, "... a distinct O-U characteristic peak appeared at the wavenumber of 1383 cm⁻¹." The peak at 1383 cm⁻¹ is already present at the infrared spectrum of the modified PPP in Figure 4(a). You should give more clear explanations for all the peaks, and you should cite all the peaks that were given in previous studies.

Response: Thank you for your suggestion. We have corrected the error in our manuscript and marked red in the revised manuscript.

This phenomenon illustrates that the structure AAUI has changed. The wave number of the O-H stretching vibration peak changed to 3428 cm⁻¹, suggesting that its contribution to surface complexation reactions cannot be ignored, The peak at 1742 cm⁻¹ correspond to C=O stretching vibration, which is obviously weakened, this indicates that the influence

of uranyl ions on the adsorbent of C=O was probably related to the complexation between them. The characteristic peak is significantly enhanced and moving to high wavenumbers at near 1383 cm^{-1} , and the peak correspond to -CH symmetric bending vibrations -CHOH suggest that complexation reaction between -CHOH and UO_2^{2+} . The reason for the above change may be due to the adsorption of uranyl ions at the active site on the surface of PPP, replacing the H^+ on the surface group of the adsorbent.

4. In lines between 27 and 32 on page 8, in the section of 4.6. Effect of pH, “When $\text{pH}>5.0$, as the alkalinity of the solution increases, the hydroxide ion concentration in the solution continuously increases, hydroxide ion and uranyl ions can form some ions with lower adsorption affinity ...” However, in Fig.10 whose caption is comparison of adsorption properties between modified and unmodified PPP, the parameters are $C_0=35\text{ mg/L}$, $V=20\text{ mL}$, $T=50^\circ\text{C}$, $m=10\text{ mg}$, $\text{pH}=6.0$, $t=90\text{ min}$. Why did you adjust pH to 6.0? Also, readers would want to see uranyl ion adsorption by the modified PPP at $\text{pH}>7.0$.

Response: Thank you for your suggestion. We have added an explanation in our manuscript and marked red in the revised manuscript.

Our study concluded that the optimal adsorption pH is a range, considering that we want to make the adsorption conditions closer to neutral. Herein, we chose $\text{pH}=6.0$

when $\text{pH}=5.0-6.0$, the adsorption rate and the amount of adsorption

change only slightly; When $\text{pH} > 6.0$, the adsorption rate and the amount of adsorption are lower. So the optimum pH for the adsorption of uranyl ions by modified PPP was 5.0-6.0.

5. There are many papers related to the adsorption of uranium ions. Therefore, authors can add the comparisons of uranium ion adsorption capacities by different adsorbents. Also, their costs and efficiencies can be added.

Response: Thank you for your suggestion. After our investigation, we found that most of their research is synthetic materials, raw materials are not easy to obtain, and the PPP materials we prepare are cheap and easy to obtain. It is well known that the production of Pomelo is extremely rich in China. Simultaneously, pomelo peel is often thrown away as waste. It can be said that the raw material adsorbent with pomelo peel as the adsorbent costs relatively low, but also reduces environmental pollution. Therefore, our research aims to make full use of a large amount of agricultural waste to provide a new material for nuclear industry wastewater treatment.

Reviewer: 3

Comments to the Author(s)

The authors used the PPP as raw materials to prepare the adsorbent. The factors of different adsorption conditions such as pH value, adsorption temperature, mass of adsorption time, and initial concentration of UO_2^{2+} were investigated. This work has a certain innovation. However, there

are several imperfections in the manuscript and the experimental results haven't presented well. Therefore, a major revision should be finished in order to suitable for these journals, and the detailed comments are as follows:

1. The introduction parts should re-write.

Response: Thank you for your suggestion. We have improved the introduction section

2. Page 7, line 10, the specific surface area, average pore diameters, and pore volume of the unmodified PPP should be added in the Table 1.

Response: Thank you for your suggestion. We have added these data of the unmodified PPP (Table 1 BET surface area , Pore volume and average Pore diameters of samples).

3. Page 8, line 46, it can be seen that when the temperature is between 50-70 °C, the adsorption performance of UO_2^{2+} is the best. Typically, in adsorption experiment, we may like to choose room temperature. The author said the best temperature is 50-70 °C. Please give the reasons.

Response: Thanks for your careful reading. We are sorry to have an error writing. we have corrected it and marked red in the revised manuscript. The thermodynamic experiments show that the adsorption processes is an endothermic reaction, when the temperature is between 50-70°C, the adsorption capacity and the adsorption rate change only slightly with the increase of temperature. Usually, the wastewater

produced by the nuclear industry has a certain temperature. For ease of operation, experiments were conducted at 50 °C.

4. Page 9, line 3, with the increase of adsorbent dose, the adsorption amount decrease from 23.715 mg/g to 6.583 mg/g. Please give lots of experimental data to prove it.

Response: Thank you for your suggestion. We are sorry to have an unclear writing.

The effect of the mass of adsorbent dose on the adsorption was analyzed. As shown in Figure 7, increasing the adsorbent dose from 250 mg/L to 1250 mg/L, the adsorption efficiency of uranyl ions gradually increased from 59.29% to 82.30%. The main reason is that with the amount of adsorbent increasing, the number of active sites for adsorption increases and the adsorption rate of uranyl ions can also gradually increase; With the increase of adsorbent dose, the adsorption amount decreases from 23.715 mg/g to 6.583 mg/g. however, the concentration of uranyl ions is constant, with the mass of adsorbent increasing, the amount of adsorbed uranium per unit mass decreases, so that the amount of adsorption decreases.

5. Page 9, line 29, the adsorption of uranyl ions by modified PPP mainly takes place on the outer surface and the inner surface of the sample. Please give the reasons.

Response: Thank you for your suggestion. We are sorry to have an

unclear writing. The adsorption mechanism is obtained by reference to some literatures, such as Chemical Engineering Journal. 2018, 343, 460-466, and references are listed in our article

6. Page 19, fig. 6 should be improved.

Response: Thank you for reading carefully. We have corrected the error and marked them red in the revised manuscript.

7. The authors should prove the recyclability of the absorbed by performing recycling experiments .

Response: Thank you for your suggestion. We have added the recycling experiments

and marked red in the revised manuscript

8. There exist some minor errors including spelling grammar and format for this paper.

Response: Thank you to give the proper word. We have corrected the error and marked them red in the revised manuscript.

Reviewer: 4

Comments to the Author(s)

The authors made a new chemical modified PPP from pomelo pulp (PPP) by fermentation, cooking and freeze-drying. The adsorbent was

characterized by means of SEM, IR, BET and EDS. The effects of different adsorption conditions (pH, adsorption temperature, adsorbent quality, adsorption time, initial concentration of UO_2^{2+}) on the adsorbent performance were studied.

The experiment of this paper is feasible, but there is no obvious innovation. There are many results and papers about similar modified adsorbents. The adsorption mechanism was discussed but not enough. The optimum adsorption conditions obtained by the author are relatively common, and there is no new knowledge about adsorption behavior. However, the PPP chosen by the authors, as a waste, can be used for other purposes rather than simply adsorption. I suggest that they have an in-depth understanding of the potential of this material and make full use of it in a more appropriate place (or valuable products, such as chemicals) instead of using complex steps to make water treatment adsorbents.

Response: Thank you for your suggestion. The purpose of our preliminary research is to provide a cheap and easy-to-obtain material for the treatment of nuclear industry wastewater. Due to the huge output of agricultural waste, such as Pomelo peel's pulp, we will make rational use of it and contribute to the environmental management. For the adsorption mechanism, we will continue In-depth study in the later stage.

We appreciate for Editors/Reviewers' warm work earnestly, and hope that the correction will meet with approval.

Once again, thank you very much for your comments and suggestions.

With best regards,

Pengfei YANG